# Spread of Infectious Disease Modeling and Analysis of Different Factors on Spread of Infectious Disease Based on Cellular Automata

**DOI:** 10.3390/ijerph16234683

**Published:** 2019-11-25

**Authors:** Sheng Bin, Gengxin Sun, Chih-Cheng Chen

**Affiliations:** 1School of Data Science and Software Engineering, Qingdao University, Qingdao 266071, China; sungengxin@qdu.edu.cn; 2Information and Engineering College, Jimei University, Xiamen 361021, China

**Keywords:** infectious disease dynamics, cellular automata, propagation model, pandemic influenza A, H1N1, numerical simulation

## Abstract

Infectious diseases are an important cause of human death. The study of the pathogenesis, spread regularity, and development trend of infectious diseases not only provides a theoretical basis for future research on infectious diseases, but also has practical guiding significance for the prevention and control of their spread. In this paper, a controlled differential equation and an objective function of infectious diseases were established by mathematical modeling. Based on cellular automata theory and a compartmental model, the SLIRDS (Susceptible-Latent-Infected-Recovered-Dead-Susceptible) model was constructed, a model which can better reflect the actual infectious process of infectious diseases. Considering the spread of disease in different populations, the model combines population density, sex ratio, and age structure to set the evolution rules of the model. Finally, on the basis of the SLIRDS model, the complex spread process of pandemic influenza A (H1N1) was simulated. The simulation results are similar to the macroscopic characteristics of pandemic influenza A (H1N1) in real life, thus the accuracy and rationality of the SLIRDS model are confirmed.

## 1. Introduction

Infectious diseases are diseases that can be transmitted from person to person, from person to animal, or from animal to animal after proto-microorganisms and parasites infect human beings or animals [1,2,3]. Infectivity, epidemic, and uncertainty are the three main characteristics of infectious diseases. A thorough study of the spread causes, spread routes, spread processes, and epidemic laws of infectious diseases is the main method for effective prevention, control, and elimination of infectious diseases.

At present, the mathematical study of infectious diseases is mainly based on the theory and method of infectious disease dynamics [4,5,6]. The essence of infectious disease dynamics is to establish a mathematical model that can reflect the spread process, spread law, and spread trend of infectious diseases. Its advantage is that, according to the characteristics of infectious diseases, the model of infectious diseases is reasonably assumed, the appropriate parameters are set, and the appropriate variables are selected. Then, the dynamic characteristics of infectious diseases can be clearly revealed. It has laid a solid foundation for further analysis of the causes and key factors of the spread of infectious diseases, and for seeking the optimal strategies for the prevention and control of such diseases.

The main method to study and forecast the spread mechanism of infectious diseases is to establish mathematical models. Some of these are used to study the general laws of infectious diseases, while others are used to study specific infectious diseases, such as HFMD (Hand foot and mouth disease), tuberculosis, AIDS, and so on. In 1760, Bernoulli [7] began using mathematical models to study the spread of smallpox by vaccination. In 1906, Hamer [8] constructed and analyzed a discrete time model for the study of recurrent measles epidemics. In 1927, Kermack and McKendrick [9] proposed the SIR (Susceptible-Infected-Recovered) compartmental model for the first time in order to study the epidemic law of the Black Death prevailing in Europe at that time. On the basis of the SIR model’s analysis, the "threshold theory" was proposed to distinguish the spread or regression of the disease. The validity of the SIR model has been proven by the data regarding large-scale infectious diseases in history, thus the deterministic model [10] based on a differential equation has been widely accepted.

With the deepening of research, the factors involved in establishing the mathematical model are increasing, and the dimension of the model is also increasing. Based on the classical SIR model, Aron and Schwartz [11] proposed the SEIR(Susceptible-Exposed-Infective-Recovered) model in 1984. That model considers that the latency of infectious diseases also has an impact on infectious diseases, so the model is more realistic. Focusing on Severe Acute Respiratory Syndrome (SARS) infectious disease spread in recent years, Safi and Gumel [12] constructed the SEQIJR(Susceptible-Exposed-Quarantine-Infective-Isolation-Recovered) model according to the characteristics of SARS. Small and Chi [13] studied the effects of vaccination and isolation on the SARS epidemic and constructed the SEIRP (Susceptible-Exposed- Infective- Recovered-Persevered) model.

In addition, some researchers start with the population structure of infectious disease spread and study the spread model of infectious disease. According to the influence of age on the spread of infectious diseases, Boklund et al. [14] proposed a model to better characterize the effect of age heterogeneity on the spread of infectious diseases. According to the difference in population, Meng et al. [15] divided the population into different groups, and they proved the global stability of disease-free equilibrium and endemic equilibrium.

With the development of artificial intelligence, the network dynamics model has gradually become a new research method of infectious disease model. The most common network dynamics models include the ordinary differential equation model, the discrete differential equation model, the impulsive differential equation model, and the differential equation model with time delay. The main methods are finite equation theory, matrix theory, bifurcation theory, K-order monotone system theory, central manifold theory, Lasalle invariant principle, etc. However, these research methods are all theoretical studies on infectious diseases, but it is difficult to apply them to practical problems.

The concept of cellular automata was proposed in the 1940s. Cellular automata can be described as a dynamic system consisting of a transformation function in cellular space, which is discrete in time and space. The cellular automata model can be formally expressed as $CA=\left(L_{C},S,M,f\right)$, where CA represents the cellular automaton system, $L_{C}$ represents the mesh space which is divided according to given rules, a mesh corresponding to a cell of cellular automata, S represents the set of cell states, $M=\left(C_{1},C_{2},\ldots ,C_{n}\right)$ represents the set of current cell adjacent to cells, and f represents the transformation function which can transform $C_{n}$ to C, the function that can calculate the state of cell C at *t* + 1 time according to the state of adjacent cells of cell C at *t* time. Adjacent cells are Moore neighbors with radius = 1. As shown in Figure 1, cells can move in eight directions.

Misra et al. [16] gave a comprehensive introduction to the theory and application of cellular automata. Since the 1990s, epidemic spread models based on cellular automata [17,18,19,20] have been extensively studied. According to the characteristics and mechanism of AIDS, Pan et al. [21] proposed a spread model of AIDS based on cellular automata. López et al. [22] proposed an epidemic spread model based on cellular automata that considers individual heterogeneity, population mobility ratio, and individual maximum moving distance. Because cellular automata can perform some experiments that cannot be done in real life by modeling, we can analyze the actual situation and obtain the results, in order to solve the complex problems that cannot be dealt with in the deterministic model. It is thus becoming a typical representative of the network dynamics model.

Based on the ability of cellular automata to model complex problems, this paper considered that, in real society, population mobility is caused by economic development, living environment, education level, and other factors, and that population density, sex ratio, and age structure of area also have some influence on the spread of infectious diseases. An epidemic spread model Susceptible-Latent-Infected-Recovered-Dead-Susceptible (SLIRDS) based on cellular automata was therefore established.

The contributions of our research are as follows:A more realistic epidemic spread model based on cellular automata was established and achieved good results in simulation experiments.The effects of population density, sex ratio, and age structure on the spread of infectious diseases were discussed, and the simulation results were analyzed to observe the effects of the above three factors on the spread process of infectious diseases.The suggestions given in this paper based on the three influencing factors provided strong support for researchers to study the spread process of infectious diseases in different environments.

The rest of this paper is organized as follows. The methodology of our research is introduced in Section 2. Simulation results and analysis are given in Section 3. Discussions are presented in Section 4. Finally, conclusions are given in Section 5.

## 2. Methodology 

### 2.1. SLIRDS Model

On the basis of the SIR and SIS (Susceptible-Infected-Susceptible) models, the state of the population was divided into susceptible, latent, infected, recovered, and dead. The total number of members of the population is denoted as N(t). S(t) represents susceptible population, meaning the number of members of the population who are not infected but are susceptible to infection at time *t*. Latent population is denoted as L(t), meaning the number of members of the population infected at time *t* but not yet affected, and at this time the individual is not infectious. Infected population is denoted as I(t), meaning the number of members of the population who are infected and have infectivity at time *t*. Recovered population is denoted as R(t), meaning the number of members of the population who are immune at *t* time and will not be infected for a certain period of time. Dead population is denoted as D(t), meaning the number of members of the population who died of infectious diseases at time *t*, and individuals are not infectious at the moment.

The SLIRDS model can be described through the following differential equation models:(1)$$\left\{ \begin{array}{c} \frac{dS\left(t\right)}{d\left(t\right)}=\delta R\left(t\right)-\beta S\left(t\right)I\left(t\right)\\ \frac{dL\left(t\right)}{d\left(t\right)}=\delta S\left(t\right)I\left(t\right)-\omega L\left(t\right)\\ \frac{dI\left(t\right)}{d\left(t\right)}=\omega L\left(t\right)-\gamma I\left(t\right)-\lambda D\left(t\right)\\ \frac{dD\left(t\right)}{d\left(t\right)}=\lambda I\left(t\right)\\ \frac{dR\left(t\right)}{d\left(t\right)}=\gamma I\left(t\right)-\delta R\left(t\right)\\ i\left(O\right)=i_{o},s\left(O\right)=s_{o}\;\;\;\;, \end{array}\right.$$
where δ represents the proportion of the population who lost immunity to the infectious disease, β is the ratio coefficient of the infection rate, ω is the latency ratio coefficient of the infectious disease, γ is the ratio coefficient of the recovered infected population, and $i_{o}$ and $s_{o}$ represent the ratio of infected and susceptible individuals in the initial population, respectively.

The transition relationships of states in the SLIRDS model are shown in Figure 2.

In order to simulate the phenomenon of crowd movement in the real world, this paper introduced the idea of random walk cellular automata to simulate individual movement in the crowd. Considering the limitation of individual movement, the maximum step length *L* is set for individual movement. At the same time, considering the individual activity *m*, all individuals are scanned randomly in each time step, and the individuals whose proportion is *m* are selected. $d_{i}$ and $d_{j}$ ($\left|d_{i}\right|,\left|d_{j}\right|\leq L$) are chosen randomly for each selected individual $C_{\left(i,j\right)}$, and then $C_{\left(i,j\right)}$ and $C_{\left(i+d_{i},j+d_{j}\right)}$ are exchanged to complete the individual movement.

In this paper, the SLIRDS epidemic model based on cellular automata is proposed. Assuming that the environment of the crowd is a regular N=n×n mesh space, a sparse matrix whose density is $\rho =C_{n}/N$ ($C_{n}$ is the set of individuals) is generated randomly. Each non-zero element of the matrix represents an effective individual. M represents the neighbor set of cellular nodes, and it uses a Moore neighbor with radius = 1. $S_{\left(i,j\right)}\left(t\right)=\left\{0,1,2,3,4\right\}$ is used to represent the cell state in the *i*-th row and the *j*-th column at time *t*. Different values represent different states as follows:

$S_{\left(i,j\right)}\left(t\right)=1$ represents susceptible state, meaning that individuals are not infected and they are immune to this infectious disease;

$S_{\left(i,j\right)}\left(t\right)=2$ represents latent state, meaning that individuals have been infected, but they do not have infectivity;

$S_{\left(i,j\right)}\left(t\right)=3$ represents infected state, meaning that individuals are infected and infectious;

$S_{\left(i,j\right)}\left(t\right)=4$ represents recovered state, meaning that individuals have recovered and acquired immunity within a certain period of time;

$S_{\left(i,j\right)}\left(t\right)=0$ represents dead state, meaning that individuals are dead and they do not have infectivity. 

Because cellular automata cannot reflect every individual’s and their neighbors’ randomness, unified parameters $T_{1}$, $T_{2}$, and $T_{3}$ are introduced, $T_{1}$ representing the maximum peak of latency time for each individual, $T_{2}$ representing the maximum peak of illness time for each individual, and $T_{3}$ representing the maximum peak of immunization time for each individual. $T_{1}\left(S_{\left(i,j\right)}\left(t\right)\right)$ represents latency time of the individual, $T_{2}\left(S_{\left(i,j\right)}\left(t\right)\right)$ represents illness time of the individual, and $T_{3}\left(S_{\left(i,j\right)}\left(t\right)\right)$ represents immunization time of the individual.

### 2.2. Influence Analysis of Different Factors on Infectious Disease Spread

Because of heterogeneity among individuals, each individual shows different resistance, infectivity, and infectious range to disease. This paper considers the effects of population density, sex ratio, and age structure on infectious disease spread in the population, and discusses the influence of different factors on infectious disease spread.

#### 2.2.1. Population Density, Sex Ratio, and Age Structure

In real life, because there are differences in climate, economy, education, and medical treatment, the population is not divided by rules like cellular automata. For example, in China, the population density in the southeast coastal areas is greater than that in the northwest. In addition, because of the different distribution of business districts, schools, and hospitals, the distribution of population in the same city is not uniform. In areas with a large population density, the distance between individuals is shorter and the spread range of individuals is wider. Individuals in the population have higher contact frequency and more neighbors around them, so their infectivity and the probability of being infected also increase.

In order to study and analyze the influence of population density on infectious disease spread, each individual is mapped into a cell in the cellular automata model. When there is no individual and the individual is in dead state in a cell, they are not infectious. In order to simulate the difference in population density, the population density can be simulated by setting the value of D(t) in the initial state. At this time, D(t) does not represent the number of dead individuals, but represents that there is no individual in the cell. In this paper, a sparse matrix was used to simulate the random distribution of population and the infectious disease spread, and then the trend of infectious disease spread under different population densities as well as the influence of different population densities on infectious disease spread were analyzed.

Due to the influence of economic development and other factors, the population ratio and age structure in different regions are also different. For example, young and middle-aged people in remote mountainous areas go to work in big cities, resulting in a large number of old and young people in the original area. In areas where labor is scarce, such as coal mines and crude oil mining areas, there is an imbalance in the proportion of men to women. Therefore, it is of great practical significance to study the influence of sex ratio and age structure on infectious disease spread.

#### 2.2.2. Individual Heterogeneity

In real life, because of different living environments, living habits, resistance levels to viruses, infectious abilities to diseases, levels of drug resistance, and spread ranges, and in order to simulate the spread mechanism of infectious diseases more accurately, it is particularly important to consider individual heterogeneity, establish an infectious disease spread model, and further analyze and predict the spread mechanism of the epidemic situation.

In this paper, the probability of infection $P_{\left(i,j\right)}\left(t\right)$ was used to describe individual heterogeneity. Individual heterogeneity is determined by the individual’s resistance to disease and the infectivity of neighbor cells.

The state of neighbor cells of cell $C_{\left(i,j\right)}$ at (*i*, *j*) can be expressed by an adjacency matrix as follows:(2)$$N_{C_{\left(i,j\right)}}=\left[\begin{array}{lll} S_{\left(i-1,j-1\right)}(t) & S_{\left(i-1,j\right)}(t) & S_{\left(i-1,j+1\right)}(t)\\ S_{\left(i,j-1\right)}(t) & S_{\left(i,j\right)}(t) & S_{\left(i,j+1\right)}(t)\\ S_{\left(i+1,j-1\right)}(t) & S_{\left(i+1,j\right)}(t) & S_{\left(i+1,j+1\right)}(t) \end{array}\right]$$

After three times spread, its adjacency matrix is defined as follows:(3)$$Q_{C_{\left(i,j\right)}}=\left[\begin{array}{lll} C_{\left(i-1,j-1\right)}(t) & C_{\left(i-1,j\right)}(t) & C_{\left(i-1,j+1\right)}(t)\\ C_{\left(i,j-1\right)}(t) & C_{\left(i,j\right)}(t) & C_{\left(i,j+1\right)}(t)\\ C_{\left(i+1,j-1\right)}(t) & C_{\left(i+1,j\right)}(t) & C_{\left(i+1,j+1\right)}(t) \end{array}\right]$$

The infection rate of cell $C_{\left(i,j\right)}$ at time *t* is defined as follows:(4)$$P_{\left(i,j\right)}\left(t\right)=\frac{a}{4}\underset{\begin{array}{c} \left(k,l\right)\in Q_{C_{\left(i,j\right)}}\\ k\neq i\;and\;i\neq j \end{array}}{\sum }\left\{P_{C_{\left(i,j\right)},C\left(k,l\right)}\left(t\right)\right\}+\frac{b}{4}\underset{\begin{array}{c} \left(k,l\right)\in Q_{C_{\left(i,j\right)}}\\ k=i\;or\;i=j \end{array}}{\sum }\left\{P_{C_{\left(i,j\right)},C\left(k,l\right)}\left(t\right)\right\}$$
where $P_{C_{\left(i,j\right)},C\left(k,l\right)}\left(t\right)$ is used to describe the probability of cell $C_{\left(i,j\right)}$ infected by cell $C_{\left(k,l\right)}$ at time *t*. 

Because the probability of infection is inversely proportional to one’s own resistance, it is proportional to the infectivity of one’s neighbors. Thus, it can be expressed as follows:(5)$$P_{C_{\left(i,j\right)},C\left(k,l\right)}\left(t\right)=\sqrt{f_{C_{\left(i,j\right)},C\left(k,l\right)}\left(1-R_{C_{\left(i,j\right)}}\right)}$$
where $f_{C_{\left(i,j\right)},C\left(k,l\right)}$ represents the infectivity of cell $C_{\left(k,l\right)}$ to cell $C_{\left(i,j\right)}$ (because of the difference in the constitution of different individuals, they have different infectivity and resistance) and $f_{C_{\left(i,j\right)},C\left(k,l\right)}$ obeys (0,1) uniform distribution [23]. $R_{C_{\left(i,j\right)}}$ represents the infectious disease resistance of cell $C_{\left(i,j\right)}$. Some diseases have different influences on different sex and age groups, that is, individual sex and age differences are also important factors affecting an individual’s resistance to disease. Thus, it can be expressed as follows:(6)$$R_{C_{\left(i,j\right)}}=\left[g_{m},g_{f}\right]\left[\begin{array}{c} f_{m}\\ f_{f} \end{array}\right]\left[y_{1},\ldots ,y_{n}\right]\left[\begin{array}{c} f_{1}\\ \ldots \\ f_{n} \end{array}\right]T_{C_{\left(i,j\right)}}$$
where $g_{m}$ and $g_{f}$ represent the proportion of males and females in the population, $f_{m}$ and $f_{f}$ represent the influence coefficient of infectious diseases on males and females, $y_{1},\ldots ,y_{n}$ are the proportion of various age groups in the population, $f_{1},\ldots ,f_{n}$ are the influence coefficient of infectious diseases on *n* groups of populations, and $T_{C_{\left(i,j\right)}}$ obeys (0,1) uniform distribution [23]. The infection probability of each individual is determined by its own resistance to infectious diseases and the infectivity of its neighbors. At each time step, the individual state is updated synchronously.

#### 2.2.3. Model Evolution Rules

According to a given population density, the sparse matrix is generated to simulate the distribution of population, and then through the age structure and sex ratio, each individual sets their attribute values. The initial state of all individuals is set to *S* = 1, the state of the infected individuals is set to *S* = 2. Individuals in cells are updated according to the following rules:

(1) When $S_{\left(i,j\right)}\left(t\right)=1$, individual infection probability $P_{\left(i,j\right)}\left(t\right)$ is calculated, and then whether the individual will be transformed into $S_{\left(i,j\right)}\left(t\right)=2$ is determined. Otherwise, $S_{\left(i,j\right)}\left(t\right)=1$. Meanwhile, individual latency time $T_{1}\left(S_{\left(i,j\right)}\left(t\right)\right)=T_{1}\left(S_{\left(i,j\right)}\left(t\right)\right)+1$.

(2) When $S_{\left(i,j\right)}\left(t\right)=2$, when $T_{1}\left(S_{\left(i,j\right)}\left(t\right)\right)<T_{1}$, $S_{\left(i,j\right)}\left(t+1\right)=2$. Otherwise, $S_{\left(i,j\right)}\left(t+1\right)=3$. Meanwhile, individual illness time $T_{2}\left(S_{\left(i,j\right)}\left(t\right)\right)=T_{2}\left(S_{\left(i,j\right)}\left(t\right)\right)+1$.

(3) When $S_{\left(i,j\right)}\left(t\right)=3$, when $T_{2}\left(S_{\left(i,j\right)}\left(t\right)\right)<T_{2}$, $S_{\left(i,j\right)}\left(t+1\right)=3$. Otherwise, the individual enters into dead state with probability λ, and $S_{\left(i,j\right)}\left(t+1\right)=0$; the rest of individuals have recovered and acquired immunity, and $S_{\left(i,j\right)}\left(t+1\right)=4$. Meanwhile, individual immunization time $T_{3}\left(S_{\left(i,j\right)}\left(t\right)\right)=T_{3}\left(S_{\left(i,j\right)}\left(t\right)\right)+1$.

(4) When $S_{\left(i,j\right)}\left(t\right)=4$, when $T_{3}\left(S_{\left(i,j\right)}\left(t\right)\right)\geq T_{3}$, individual immunity to the infectious disease disappears with probability δ. The individual then turns into susceptible state, $S_{\left(i,j\right)}\left(t\right)=1$.

(5) At each time step, all individuals move.

## 3. Simulation Results and Analysis

Without considering other factors, this paper focused on the influence of three factors, namely, population density, individual heterogeneity, and mobility on infectious disease spread, and the SLIRDS model based on cellular automata was constructed. In order to verify the validity of the model, this paper took pandemic influenza A (H1N1) as an example to simulate the spread process of pandemic influenza A (H1N1).

In this paper, we used MATLAB simulation software (R2013b, MathWorks, Natick, MA, USA) to carry out 200 simulation experiments; the simulated curves are realizations of the average from all simulations. According to the latent and infectious characteristics of pandemic influenza A (H1N1), the time step of simulation is in days, and the total time is set to T = 40.

The simulated initial number of members of the infected population was consistent with the actual number of members of the infected population, and we assumed that the proportion of the initial latent population was 0.15%.

First, the number of members of the infected population in the SLIRDS model simulation experiments was compared with the actual data of pandemic influenza A (H1N1) in Beijing in Mainland China (June–July 2009) [24]. The comparison results are shown in Figure 3.

In Figure 3, the abscissa is the time step of simulation and the ordinate is the number of infected individuals. The correlation coefficient of the two sets of data is 0.97215 by *t*-test. It shows that the simulation results are close to the actual data and that the model is reasonable and effective.

### 3.1. Influence of Population Density on Infectious Disease Spread

All things being equal, the parameters of two simulations for the SLIRDS model were set as follows:

(1) Population density ρ=1, susceptible population:latent population:infected population = 0.872:0.107:0.021.

(2) Population density ρ=0.8, susceptible population:latent population:infected population = 0.6985:0.0855:0.016.

Two simulation results are shown in Figure 4.

In Figure 4, considering the difference in population base, the number of members of the population that died, were susceptible, were infected, and were immunized was replaced by death, susceptibility, infection and immunization rates to describe the changes in population in different states.

From the death rate curve in Figure 4a, it can be seen that the death rate increases with the increase of population density, but the overall trend is rising and tending to be stable.

From the susceptibility rate curve in Figure 4b, it can be seen that the change in population density has little influence on the susceptible population, and the susceptibility will first decrease and then reach a stable value when the population density is large.

From the infection rate curve in Figure 4c, it can be seen that the change in population density has little influence on the infected population. When the population density is large, the number of members of the infected population is greater, but the general trend is rising first, then falling, and finally tends to be stable.

From the immunization rate curve in Figure 4d, it can be seen that the change in population density has little influence on the immune population. When the population density is large, the immunity first rises and then reaches a stable value.

According to the above analysis, it is known that when the population density is large, the spread rate of infectious diseases is faster.

### 3.2. Influence of Sex Ratio on Infectious Disease Spread

#### 3.2.1. Influence of Infectious Disease Spread under Different Coefficients

All things being equal, the ratio of males to females was 4:1. The parameters of two simulations for the SLIRDS model were set as follows:

(1) The influence coefficients of infectious disease on males and females were 0.1 and 0.9, respectively.

(2) The influence coefficients of infectious disease on males and females were 0.9 and 0.1, respectively.

Two simulation results are shown in Figure 5.

As shown in Figure 5a, we can see that when infectious diseases have a greater influence on males, the number of deaths is higher, but the overall trend is rising and gradually stable.

As shown in Figure 5b, we can see that infectious diseases have less influence on susceptible population under different influence coefficients; when infectious diseases have a greater influence on females, the number of members of the susceptible population first decreases and then reaches a stable value.

As shown in Figure 5c, we can see that infectious diseases have less influence on infected population under different influence coefficients; when infectious diseases have a greater influence on males, the number of members of the susceptible population first decreases and then reaches a stable value. However, the overall trend is first rising and then falling, and finally tends to be stable.

As shown in Figure 5d, we can see that infectious diseases have less influence on recovered population under different influence coefficients; when infectious diseases have greater influence on males, the number of members of the susceptible population increases first, and then reaches a stable value.

According to the above analysis, it is known that in cities with more males than females, when the infectious disease has a great influence on males, infectious diseases have a greater influence on the population because of the large population base of males. Similarly, there are corresponding phenomena in cities with more females than males.

#### 3.2.2. Influence of Infectious Disease Spread under Different Sex Ratios

According to related materials [24], the influence coefficient of pandemic influenza A (H1N1) on males and females is very different. In simulation experiments, the influence coefficients of pandemic influenza A (H1N1) on males and females were set to $g_{m}=0.55,g_{f}=0.35$, respectively. The parameters of two simulations for the SLIRDS model were set as follows:

(1) The ratio of males to females is 4:1.

(2) The ratio of males to females is 1:4.

Two simulation results are shown in Figure 6.

From the death curve in Figure 6a, we can see that when the number of males is large, the number of deaths is higher, but the overall trend is rising and gradually stable. 

From the susceptibility curve in Figure 6b, we can see that under different sex ratios, infectious diseases have little influence on susceptible population. When the number of females is large, the number of members of the susceptible population first decreases and then reaches a stable value.

From the infection curve in Figure 6c, we can see that under different sex ratios, infectious diseases have little influence on the infected population. When the number of males is large, the number of members of the infected population decreases first and then reaches a stable value.

From the immunization curve in Figure 6d, we can see that under different sex ratios, infectious diseases have little influence on the immunization population. When the number of males is large, the number of members of the immunization population increases first, and then reaches a stable value. 

According to the above analysis, it is known that when the number of males is larger in the cities where infectious diseases affect men more, infectious diseases have a greater influence on the population.

### 3.3. Influence of Age Structure on Infectious Disease Spread

Due to factors such as mobility and spatial environment, age structure of the population presents different distributions. The age structure of a city can be divided into three types: young, adult, and aged according to the proportion of children, adolescents, youth, middle-aged people, and elderly people.

According to related materials [24], the influence coefficient of pandemic influenza A (H1N1) on children, adolescents, youth, middle-aged people, and elderly people is very different. In simulation experiments, the influence coefficients of pandemic influenza A (H1N1) on children, adolescents, youth, middle-aged people, and elderly people were set to $f_{1}=0.65,\;f_{2}=0.58,\;f_{3}=0.46,\;f_{4}=0.37,\;f_{5}=0.68$, respectively.

All things being equal, the parameters of three simulations for the SLIRDS model were set as follows:

(1) Young: children: adolescents: youth: middle age: old age = 0.025:0.05:0.5:0.4:0.025.

(2) Adult: children: adolescents: youth: middle age: old age = 0.2:0.2:0.3:0.25:0.15.

(3) Aged: children: adolescents: youth: middle age: old age = 0.1:0.05:0.05:0.2:0.6.

Three simulation results are shown in Figure 7.

The death, susceptibility, infection, and immunization curves are shown in Figure 7a–d, respectively. It can be seen that the number of deaths in the aged cities is the largest. The number of young urban deaths is only inferior to that of the aged cities, whereas the number of deaths in the adult cities is the least. However, the overall trend of change is gradually stable after rising for all types of cities. The difference in age structure of the population has little influence on the susceptible population, and the number of members of the susceptible population in adult cities decreases first and then reaches a stable value. The difference in age structure of the population has little influence on the infected population. The number of members of the infected population in the aged cities is the highest, but the general trend is rising first and then decreasing for all types of cities. The difference in age structure of the population has little influence on the immunization population, and the number of immune individuals in the adult city rises first to then reach a stable level.

According to the above analysis, it is known that infectious diseases spread more slowly in adult cities than in aged and young cities, but the resistance of young cities to infectious diseases is slightly greater than that of aged cities.

## 4. Discussion

In this paper, we used the idea of a sparse matrix to add population density, sex ratio, and age structure factors into the SLIRDS model. Population density was set to 1 and 0.8, respectively. All things being equal, with the increase of population density, infectious diseases spread faster, and infectious diseases have a greater influence on the population. When analyzing the influence of sex ratio on the spread of infectious diseases, we considered two factors, namely, different influence coefficient and different sex ratio. First, the ratio of males to females was set to 4:1. Because of the large population base of males, infectious diseases have a greater influence on the population when the infection coefficient is greater. Second, the influence coefficients of infectious diseases on males and females were 0.9 and 0.2, respectively. Because infectious diseases have a greater influence on males, when the number of males is larger, the influence of infectious diseases on the population is greater. When analyzing the influence of age structure on the spread of infectious diseases, we simulated three types of population distribution structure, namely, young, adult, and aged, according to the age structure distribution ratio. The number of members of the infected population and deaths in the aged cities were the largest, and the susceptibility of adult cities to infectious diseases was stronger. That is, the uniform distribution of age plays a more active role in the spread of infectious diseases.

In order to effectively prevent the spread of infectious diseases in the population, we offer three suggestions according to the three influencing factors. (1) Population density: the regional economy should be balanced, the large-scale turnover of personnel should be reduced, the density of urban population should be controlled, the population in densely populated areas such as schools should be evacuated during the epidemic period of infectious diseases. (2) Sex ratio: when infectious disease has a greater influence on a certain sex, or if the sex ratio is larger in the population, attention should be paid to prevention and treatment with respect to that sex. (3) Age structure: the age structure should be optimized and the age structure of the city should be stabilized. On this basis, we should pay attention to prevention and treatment with respect to disadvantaged groups (such as the elderly and children) in the spread of infectious diseases.

Many factors affect the spread of infectious diseases. This paper only studied the influence of the above three factors on the spread of infectious diseases. The many factors that must be further explored in the future include the following: first, the influence of population activity on the spread of infectious diseases; second, the influence of population size on the spread of infectious diseases; and third, in view of the analysis of the influence factors, how to implement effective prevention and control measures against the spread of infectious diseases in specific cities.

## 5. Conclusions

In order to study the main factors that affect the spread process of infectious diseases, the SLIRDS model was proposed in this paper. Combined with cellular automata, an epidemic model based on cellular automata was established. In the simulation experiment, the influence of population density, sex ratio, and age structure on infectious disease spread was analyzed by comparing the results with those from the actual spread process of pandemic influenza A (H1N1), and the accuracy of the SLIRDS model was confirmed.

With research on the spread of infectious diseases, the advantage of using cellular automata to model complex problems can be used to optimize epidemic models. The system can better analyze the factors affecting the spread of infectious diseases, and provide better theoretical support for the prevention and control of infectious diseases.

Because cellular automata cannot reflect every individual’s and their neighbors’ randomness, there was a lack of individual randomness in the SLIRDS model for the maximum peak of each state for the different durations. This will be the direction that we take in the future to focus on improvement.

## Figures and Tables

**Figure 1 ijerph-16-04683-f001:**
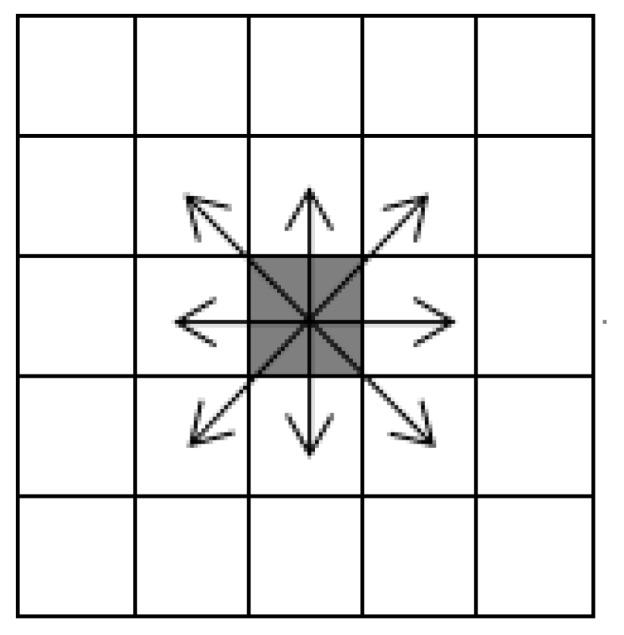
Moore neighbor (radius = 1).

**Figure 2 ijerph-16-04683-f002:**
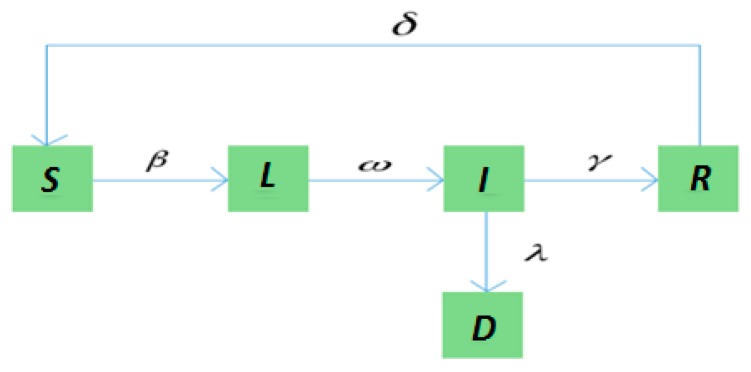
The state transition diagram of the Susceptible-Latent-Infected-Recovered-Dead-Susceptible (SLIRDS) model.

**Figure 3 ijerph-16-04683-f003:**
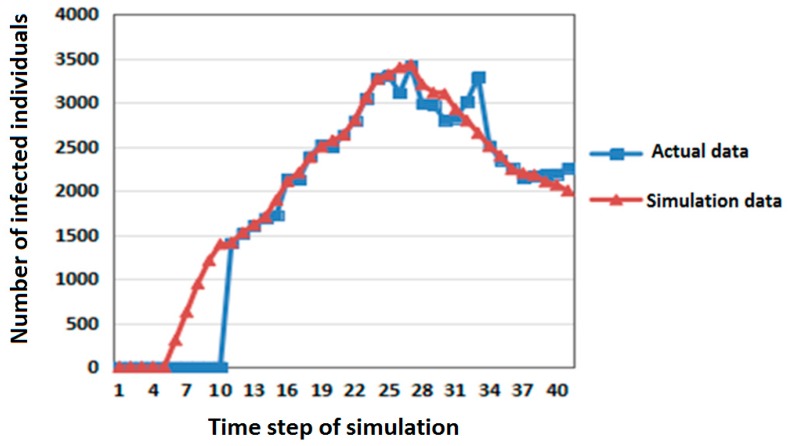
Comparison of actual data and simulation data for pandemic influenza A (H1N1) in Beijing in Mainland China (June–July 2009).

**Figure 4 ijerph-16-04683-f004:**
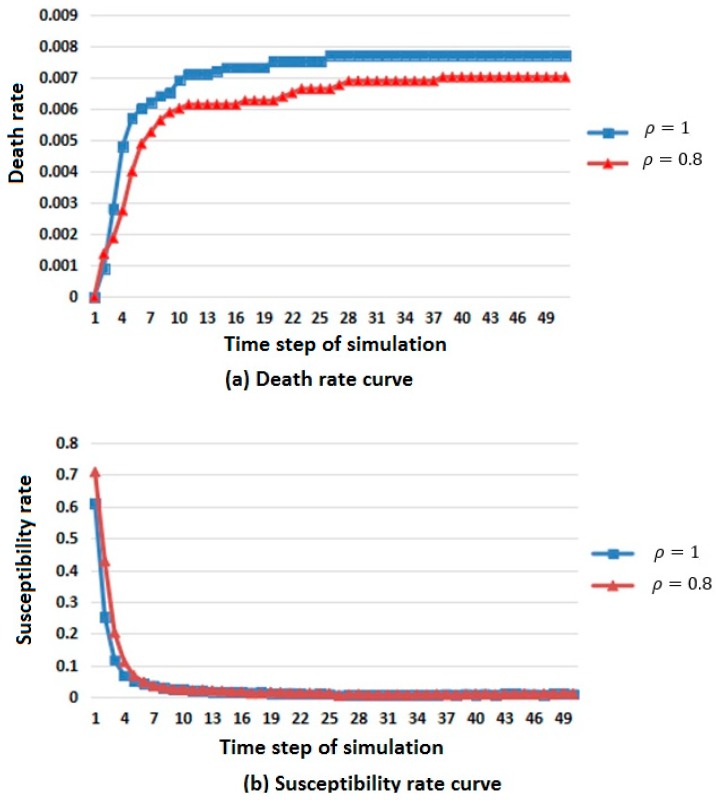
Infectious disease spread state distribution with different population density. (**a**) Death rate curve; (**b**) Susceptibility rate curve; (**c**) Infection rate curve; (**d**) Immunization rate curve.

**Figure 5 ijerph-16-04683-f005:**
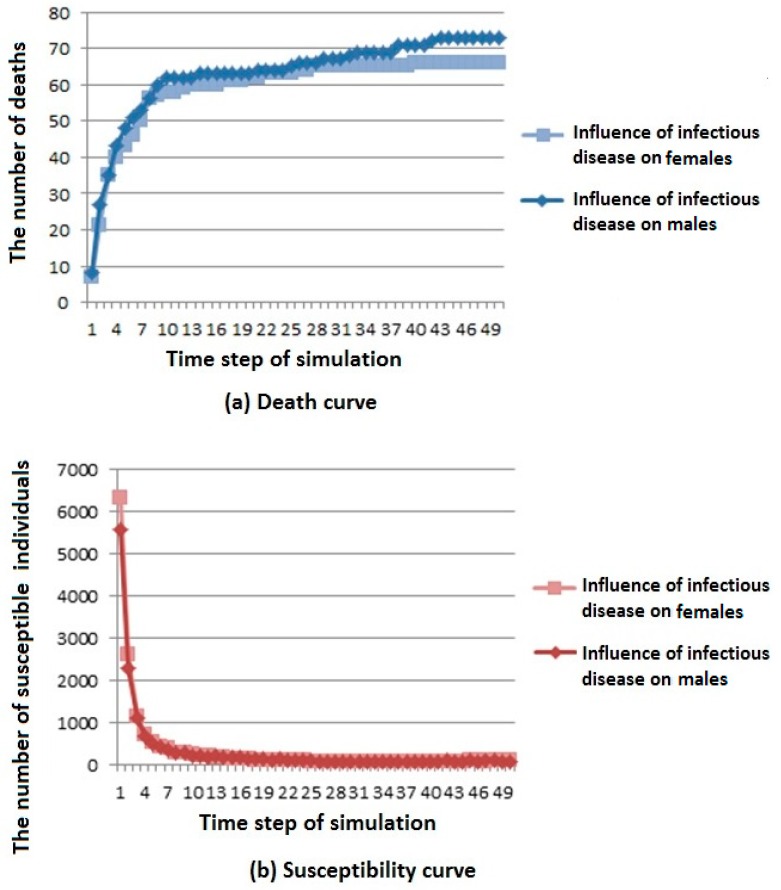
Infectious disease spread state distribution with different coefficients. (**a**) Death rate curve; (**b**) Susceptibility rate curve; (**c**) Infection rate curve; (**d**) Immunization rate curve.

**Figure 6 ijerph-16-04683-f006:**
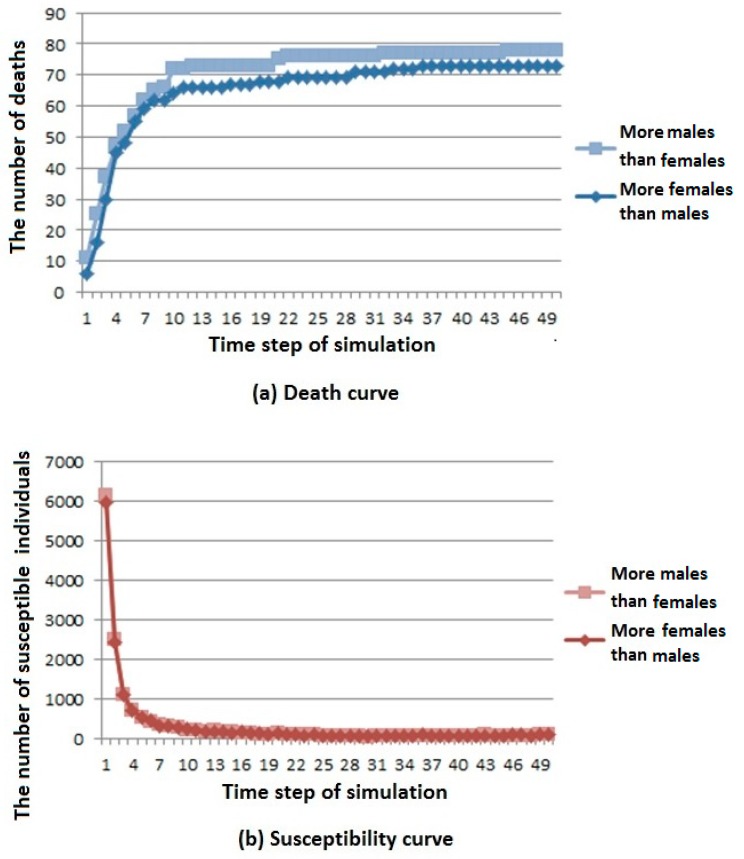
Infectious disease spread state distribution with different sex ratios. (**a**) Death rate curve; (**b**) Susceptibility rate curve; (**c**) Infection rate curve; (**d**) Immunization rate curve.

**Figure 7 ijerph-16-04683-f007:**
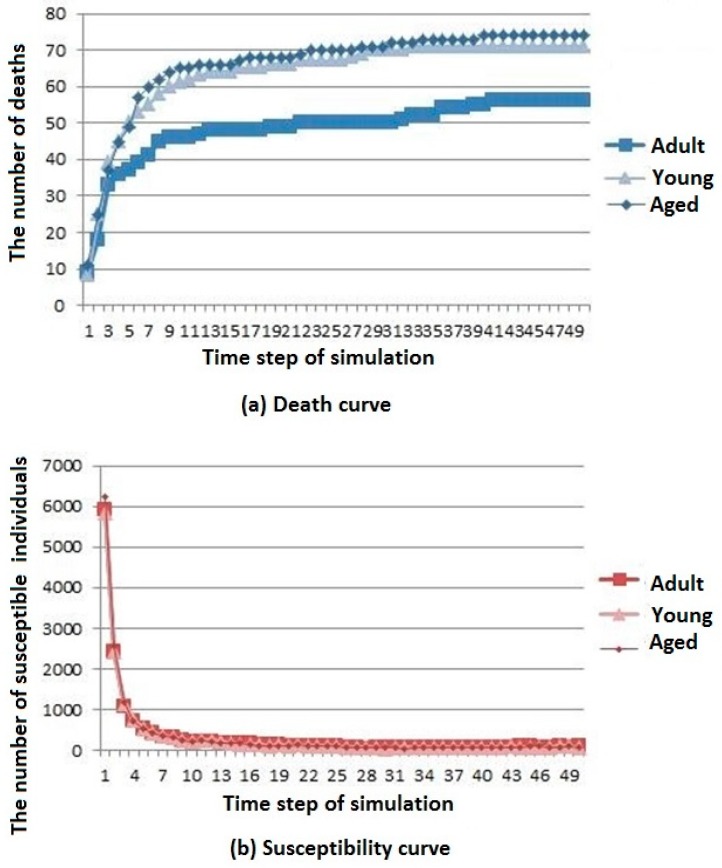
Infectious disease spread state distribution with different age structure. (**a**) Death rate curve; (**b**) Susceptibility rate curve; (**c**) Infection rate curve; (**d**) Immunization rate curve.

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
