# Peer review of "Spread of Infectious Disease Modeling and Analysis of Different Factors on Spread of Infectious Disease Based on Cellular Automata"

_ijerph, 2019, doi:10.3390/ijerph16234683_

Round 1
Reviewer 1 Report
This manuscript described how cellular automata models can be adapted to model infectious disease transmission over time in certain population, and performed several simulation studies to investigate the potential impacts from different population parameters on the disease outcome.
I have two major concerns:
Firstly, the simulation model described in the paper was quite detailed, however several model assumptions may not necessarily be validated. For example, the author introduced maximum peak of latency, illness and immunization time for each individual, but the model evolution rule indicated these time intervals were fixed at maximum. There was lack of individual randomness in these durations. Several random parameters were designed to follow uniform distribution but not explained why. Some language need to be improved. In row 201, parameters a/b seem to describe the different connection strength between two neighbouring cells whose connection are diagonal/straight, but no more discussion on how these matter in the paper. In row 122, the differential equation for dD(t)/dt was incorrect, the right side should be \lambda I(t). Same for that for dI(t)/dt. In row 130-136, it's hard to understand the paragraph. Overall the method description needs improvement.
Secondly, the data analysis and simulation part is seriously lacking. It's not clear in the figures the simulated curves are realizations of one single simulation or the average from many repeated simulations. There was no output to evaluate the variability for results in each scenario. The H1N1 data analysis is lacking details on how model parameters were chosen, and why the parameters/results are similar to what actually happened. In the simulation studies presented, the parameter scenarios used were to illustrate the difference in results instead of to reflect the reality where disease transmission occurs. For example, gender ratio of 1:4 is highly unlikely for most places. Age distribution must be combined with heterogeneity in susceptibility of different age groups, which was not mentioned in the paper. Furthermore, going back to the model, the only difference in gender and age factor is the number of subgroups modeled, otherwise these two were interchangeable. To model a disease such as HIV or influenza, very different scenarios might be used to reflect the natural disease history. My suggestion is to focus on one disease of interest and refer previous literature on how to model these parameters.
Author Response
Dear Reviewer ,
Thank you very much for your letter and for the comments by the reviewers. These comments are very valuable and helpful for our paper.
We appreciate the careful, constructive, and generally favorable reviews given to our paper by the reviewers.
We believe we have adequately addressed all the excellent advices and questions raised by reviewers. Furthermore, we checked the manuscript and made sure the submitted manuscript is correct.
Response to Reviewer 1 Comments
Point 1: Firstly, the simulation model described in the paper was quite detailed, however several model assumptions may not necessarily be validated. For example, the author introduced maximum peak of latency, illness and immunization time for each individual, but the model evolution rule indicated these time intervals were fixed at maximum. There was lack of individual randomness in these durations. Several random parameters were designed to follow uniform distribution but not explained why. Some language need to be improved. In row 201, parameters a/b seem to describe the different connection strength between two neighbouring cells whose connection are diagonal/straight, but no more discussion on how these matter in the paper. In row 122, the differential equation for dD(t)/dt was incorrect, the right side should be \lambda I(t). Same for that for dI(t)/dt. In row 130-136, it's hard to understand the paragraph. Overall the method description needs improvement. 

Response 1: Thanks for the comment. Your comments are of great help to us, we have revised the paper according to your comments.
Because cellular automata can't reflect every individual and their neighbors’ randomness, in the SLIRDS model there was lack of individual randomness for the maximum peak of each state in these durations. It will be the direction for us to focus on improvement in the future.
According to literature [23] (Gao B J, Zhang T, Xuan H Y, et al. A Heterogeneous Cellular Automata Model for SARS Transmission[J]. Journal of Systems & Management, 2006, 15(3):205-209.), several random parameters were designed to follow uniform distribution.
In row 201, parameters a/b is used to describe the different connection strength between two neighbouring cells, they are reflected in the simulation experiment.
The mistakes in row 122 had been revised.
The content in row 130-136 is used to describe the random walk process in cellular automata.
Point 2: Secondly, the data analysis and simulation part is seriously lacking. It's not clear in the figures the simulated curves are realizations of one single simulation or the average from many repeated simulations. There was no output to evaluate the variability for results in each scenario. The H1N1 data analysis is lacking details on how model parameters were chosen, and why the parameters/results are similar to what actually happened. In the simulation studies presented, the parameter scenarios used were to illustrate the difference in results instead of to reflect the reality where disease transmission occurs. For example, gender ratio of 1:4 is highly unlikely for most places. Age distribution must be combined with heterogeneity in susceptibility of different age groups, which was not mentioned in the paper. Furthermore, going back to the model, the only difference in gender and age factor is the number of subgroups modeled, otherwise these two were interchangeable. To model a disease such as HIV or influenza, very different scenarios might be used to reflect the natural disease history. My suggestion is to focus on one disease of interest and refer previous literature on how to model these parameters.
Response 2: Thanks for the comment.
In this paper, we used Matlab simulation software to carry out 200 simulation experiments, the simulated curves are realizations of average from all simulations. According to the latent and infectious characteristics of the pandemic influenza A (H1N1), the time step of simulation is in days, and the total time is set T = 40.
The H1N1 data analysis is comes from Beijing (June 2009 - July 2009), model parameters had been given detailed.
As you said, gender ratio of 1:4 is highly unlikely for most places. We set this extreme ratio just to better express the impact of sex ratio on infectious diseases.
In our paper, we think individual gender and age differences are also important factors affecting individual's resistance to disease. So we can the equation as follows:
Where and represent influence coefficient of infectious diseases on male and female, are the influence coefficient of infectious diseases on n groups of populations.
In our simulation experiments, , and had been set appropriate values according to refer previous literature. In the revised version, we had given these parameter values.

Reviewer 2 Report
The subject of the paper is interesting. The introduction provides sufficient information about the content. Although simulation is made in a relative small group the paper seems to show a small novelty. However some moderate english changes should be applied. I would recommend to give it to the native speaker.
Also some minor changes should be done to improve the paper:
In 175-177 the sentence is repeated.
In all figures the axis description is missing. Also the results are present with one pattern. It should be written in more scientific way.
Line 275 it should be 4d not, 4c.
Discussion should be written more precisely e.g. sentence in line 389 is missing sense.
Author Response
Cover Letter
Dear Reviewer ,
Thank you very much for your letter and for the comments by the reviewers. These comments are very valuable and helpful for our paper.
We appreciate the careful, constructive, and generally favorable reviews given to our paper by the reviewers.
We believe we have adequately addressed all the excellent advices and questions raised by reviewers. Furthermore, we checked the manuscript and made sure the submitted manuscript is correct.
Response to Reviewer 2 Comments
Point 1: In 175-177 the sentence is repeated. 

Response 1: Thanks for the comment. Duplicate sentence had been deleted.
Point 2: In all figures the axis description is missing. Also the results are present with one pattern. It should be written in more scientific way.
Response 2: Thanks for the comment. In all figures the axis description had been added. And the results were present by different description way.
Point 3: Line 275 it should be 4d not, 4c.
Response 3: Thanks for the comment. We have corrected this mistake.
Point 4: Discussion should be written more precisely e.g. sentence in line 389 is missing sense.
Response 4: Thanks for the comment. The wrong sentence has been deleted.
Round 2
Reviewer 2 Report
I am ready to accept the paper in his current state.